# Type and distribution of sensilla in the antennae of *Euplatypus parallelus* (F.) (Coleoptera: Curculionidea, Platypodinae)

Gaoke Lei[1], Yueguan Fu[2], Wei-Jian Wu[1]*

1 Laboratory of Insect Ecology, South China Agricultural University, Guangzhou, China, 2 Environment and Plant Protection Institute, Chinese Academy of Tropical Agricultural Science, Haikou, China

* weijwu@scau.edu.cn

**Data Availability Statement:** All relevant data are within the manuscript and its Supporting Information files

**Funding:** This study was supported by the Special Fund for Agro-scientific Research in the Public

## Abstract

*Euplatypus parallelus* (F.) (Coleoptera: Curculionidea) is the most destructive cosmopolitan insect pest of the Platypodinae. Pheromone-based luring agents are used currently in controlling bark beetle. Antennae are the primary insect organs sensing volatiles of host trees and pheromones of pioneer males. We studied the external morphology of antennae and the type, distribution, and the number of the beetle sensilla. Our results show *E. parallelus* have a geniculate antenna composed of 6 segments, namely the scape, 4-segmented funicle and club. Ninety-seven percent of the antennal sensors were distributed in the club, and 3% were distributed in the scape and funicle. 6 types of sensilla on the antennae were found, including sensilla trichodea (subtypes: STI, STII and STIII), sensilla basiconica (subtypes: SBI, SBII, SBIII and SBIV), sensilla chaetica (subtypes: SChI, SChII and SChIII), as well as sensilla coeloconica, sensilla campaniform and sensilla furcatea. There was no significant difference in the type, distribution and number of sensilla in males and females. No significant difference in the shape and distribution of antennae was found between sexes, but the length of antennae and the number of SChI, SChII, STI, SBI, SBIII and SBIV were significantly larger in females than males. We revealed the external cuticular structure of the antennae in *E. parallelus*, which can be used to guide future electrophysiological investigations to understand the ability of this beetle to detect semiochemicals.

## Introduction

Platypodinae is one of the most important groups of forest pests that damage mainly weakened or felled coniferous or broad-leaved trees with a diameter breast height (DBH) >20 cm [1]. They penetrate the xylem and oviposit in their host trees [2], weakening the trunk and causing it to break under extreme conditions, eventually resulting in trees wilting and dying [3, 4]. Some aggressive species of Platypodinae can also endanger living standing trees, which is a threat to the forest eco-systems in many areas of the world [5].

*Euplatypus parallelus* (F.) is the most destructive cosmopolitan insect pests of the Platypodinae [6–8]. This beetle has its origin in Central and South America, but its current distribution

Interest of China (201103026-4) and the Earmarked Fund for China Agriculture Research System (CARS-34-BC2). The funders had no role in study design, data collection and analysis, decision to publish, or preparation of the manuscript

**Competing interests:** The authors have declared that no competing interests exist

includes Madagascar, Australia, Africa, and Southeast Asia due to the transport of infested timber between countries [9, 10]. From 2016 to 2017, Li et al. [11] hand collected and captured *E. parallelus* in light traps from weakened trees in Danzhou, Hainan, which was only the first record of this beetle in China. This beetle is a notably polyphagous pest, it has been reported already on more than 80 different host-tree species in 25 distinct families, such as *Eucalyptus*, rubber tree, pine, etc [12, 13]. Stressed or weakened trees are particularly subject to attack by *E. parallelus*. Some healthy trees were also damaged by this beetle, and it even can breed in thin trunks of about 10 cm in diameter [9]. Pioneer males of *E. parallelus* uses mainly olfactory cues to locate host plants. They release a pheromone to attract other males and females, leading to mass-attack [14]. Numerous tunnels are burrowed in the tree trunk by this beetle. Through the female beetle's mycangia, the ambrosia fungus is plural into the galleries. The bark tunnels with the associated symbiotic fungus can greatly reduce the value of the timber. Fungi are the nutritional sources of adults and larvae [9].

*E. parallelus* is difficult to control with pesticides because this beetle is small and propagates rapidly, and it lives hidden away. Platypodinae have few natural enemies. Silva et al. [15] found Colydiidae and Trypanaeus in tunnels of *E. parallelus*, which are known Platypodinae predators, but they might not be able to control the rapid propagation of bark beetles. Claus and Gary [16] found that host plant volatiles and pheromones play an important role in host-location and mass-attack by *E. parallelus*. Synthetic pheromones are a common method for controlling bark beetles. They are utilized for population monitoring, mating disruption and mass-capturing [17].

Semiochemicals are essential for activities such as survival, reproduction and host seeking by *E. parallelus*. This beetle senses volatiles of host trees and pheromones of pioneer males mainly by using antennae [18, 19]. The antennae are the main external sensory organs in bark beetles; they have a variety of sensory organs and serve different sensory modalities, having the functions of smell, touch, temperature, taste, and humidity [20]. The antennae receive chemical communication [21, 22]. There is little research has involved the sensilla in the antennae of Platypodinae. However, various sensilla have been researched regarding the function, external cuticular structure and morphology of Scolytinae, close relatives of Platypodinae, such as *Dendroctonus ponderosae* Hopkins [23], *Xylosandrus germanus* Blandford, *Xylosandrus crassiusculus* [24], *Xylosandrus compactus* [25], and *Ips acuminatus* Gy11 [26]. This work contributed significantly to understanding sensilla in the antennae of Platypodinae. In the present study, we used FESEM to describe and analyze the morphology, structure, distribution, and quantity of the antennal sensilla in *E. parallelus*, which will provide a theoretical basis for revealing the host recognition mechanism in *E. parallelus*.

## Materials and methods

### Insects

Sections of the main stem were cut from the rubber trees (*Hevea brasiliensis*) infested by *E. parallelus*. Sample collection was conducted in January 2019 at the Xiqing farm, in Danzhou, Hainan province (19°31′N, 109°34′E). The cut ends of the logs (approximately 1 m in length) were sealed with paraffin to minimize water loss. The adult beetles were collected after emerging and transferred into plastic boxes. In discriminating sex, the method of Atkinson [27] and Wood [28] was used. The beetles were then preserved in 75% ethanol at 4°C for future study.

### Field emission scanning electron microscopy

An FESEM was used to observe 12 males and 12 females of *E. parallelus*. The antennae were excised under 80×magnification (Carl Zeiss Microimaging GmbH 37081 Göttingen,

Germany). The specimens were placed inside a tube with 75% ethanol. The antennae were cleaned for two minutes with an ultrasonic wave cleaner. This treatment was repeated five times. After natural drying for 24 h, the treated antennae were fixed on a stub with adhesive tape. Finally, they were coated with gold-palladium and photographs were acquired using an FESEM operated at 3 KV (Verios 460, FEI, Czech Republic).

## Data analysis

All antennal properties were measured in at least 10 females and 10 males using a slide caliper (GB/T1214.1–1214.4). Length was measured from the tip to the base midpoint of the sensilla, and width was measured at the bottom of the sensilla. All the data were analyzed by SPSS 23.0 software (http://www.spss.com). Differences between the data were determined by the Mann-Whitney *U* test.

## Terminology

The terminology used in this study is based on that used by Schneider [20], Hallem et al. [29], Keil et al. [30] and Wang et al. [31].

## Results

### Antennal morphology

It was observed by FESEM that there is same difference in antennae morphology between males and females. The base of antennae was jointed proximally to the compound eye, was curved slightly inward. Both *E. parallelus* male and female adults had a geniculate antenna composed of three segments: a scape, 4-segmented funicle and a club (Figs 1A and 2A). The antennae were significantly longer in females than males (Table 1). The first antennal segment (scape) is large and long with a depression at the junction of the head. The base was curved in a "U" shape, and the surface had longitudinal furrows (Figs 1D and 2C). Four antennomeres (F1-F4) composed the funicle, with the surface having furrows. The 1st funicular antennomere (F1) was swollen, about half of it was embedded in the scape. The 2nd funicular antennomere (F2) was the thinnest and gradually widened in subsequent funicular antennomeres. The last funicular antennomere (F4) was jointed to the club (Figs 1C and 2D). The terminal club was broad and flattened, exhibiting an approximate oval-shaped region that contained most of sensilla (Figs 1B and 2B). The width of the scape, 4-segmented funicle and the club were significantly larger in females than males. In addition, the length of the club in females was significantly longer than in males (Table 1).

### Sensilla types

Based on morphological characteristics, we identified various types of sensilla in *E. parallelus*. The antennal sensilla in females and males were divided into 6 types and 13 structure subtypes, including sensilla trichodea, sensilla basiconica, sensilla chaetica, as well as sensilla coeloconica, sensilla campaniforme and sensilla furcatea. No significant difference in terms of sensilla types was found between sexes (Table 2). The number of sensilla in the club of female and male adults *E. parallelus* was the largest, followed by the scape, and the funicle was the least. The number of sensilla in the club and scape was significantly more in female adults than male adults (Table 3).

### Sensilla morphology and structure

Sensilla trichodea were without any specialized basal cuticular ring serving as articulating membrane. ST appeared dispersed in the club, without a clear pattern of distribution. Based

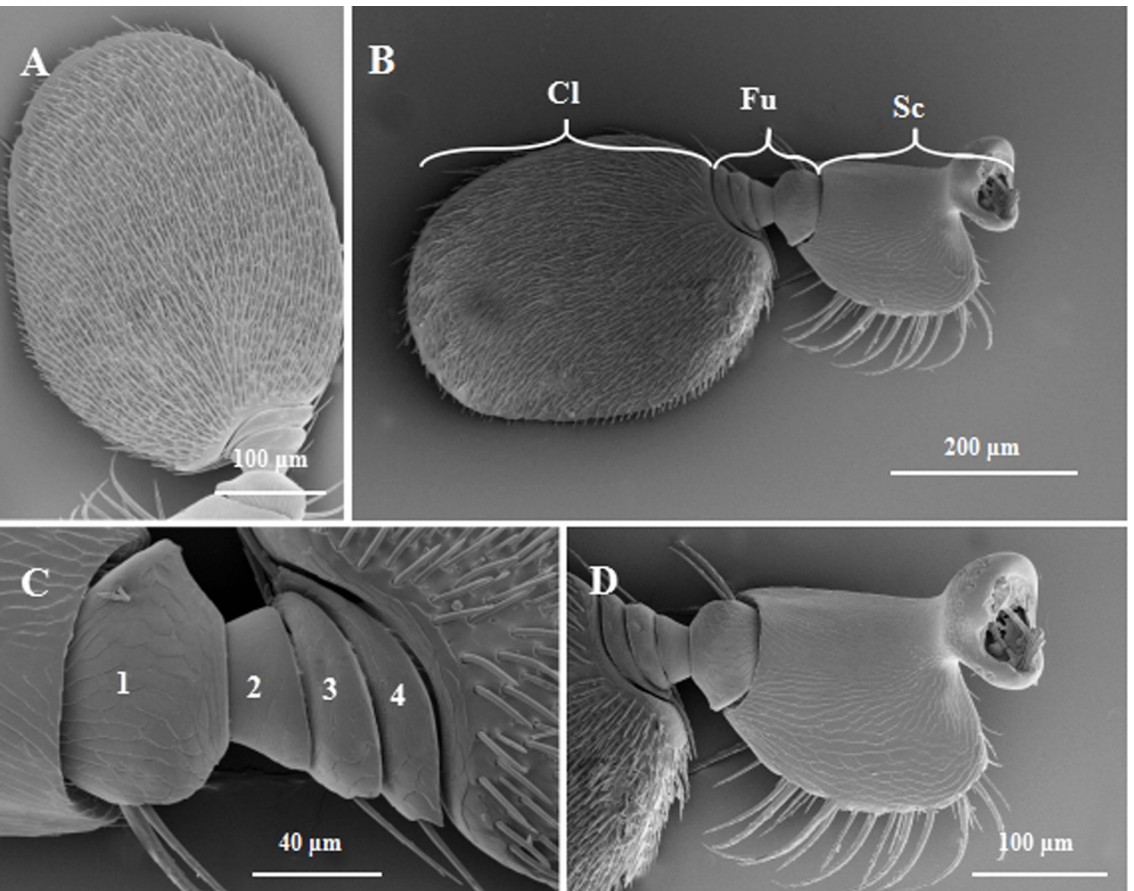

**Fig 1. Adult *Euplatypus parallelus* antennae in dorsal vision.** *E. parallelus* have a geniculate antenna, is sensory appendage on either side of the rostrum, and composed of 6 segments, namely the scape, 4-segmented funicle, and club. The antennae of females are significantly longer than males. A: Geniculated antennae of E. parallelus. B: Club. C: Funicle. D: Scape. Cl: Club; Fu: Funicle; Sc: Scape.

on their morphological and size features, we distinguished three subtypes of ST: I, II and III. STI had a wider base. It was straight or slightly curved with a sharp tip, its wall was smooth and multiporous (Fig 3A). It's number was more larger in females than males (Table 4). STII was similar to STI, but it was longer than STI (Fig 3B). This sensillum (STII) had a sharper tip than STI. There were more pores in the wall of STII than STI. STIInumber were more larger than STI in both sexes (Table 4). STIII was trichoid (Fig 3C). This sensillum had a sharper tip than STI. There were fewer pores in the wall of STIII than STI, tapering gradually from the base to the end. The number of STIII in antennae was less than that of STI. No significant differences in the number of STII and STIII were found between males and females (Table 4).

Sensilla chaetica were shaped like a thorn. Their wall was smooth, but they lacked pores, they were immersed in a deep socket, tapering gradually from the base to the end. SCh distribution in scape, funicle and club differed. This type was the most widely distributed structure on the antennae. Based on their morphological structure, we distinguished three subtypes of SCh: I, II and III. SChI was straight or slightly curved and longer than ST (Fig 3D). Its base was jointed within a socket. There were long longitudinal furrows in the wall. There were about 5 spine-like branches in the wall of this type. The number of SChI in the club was the

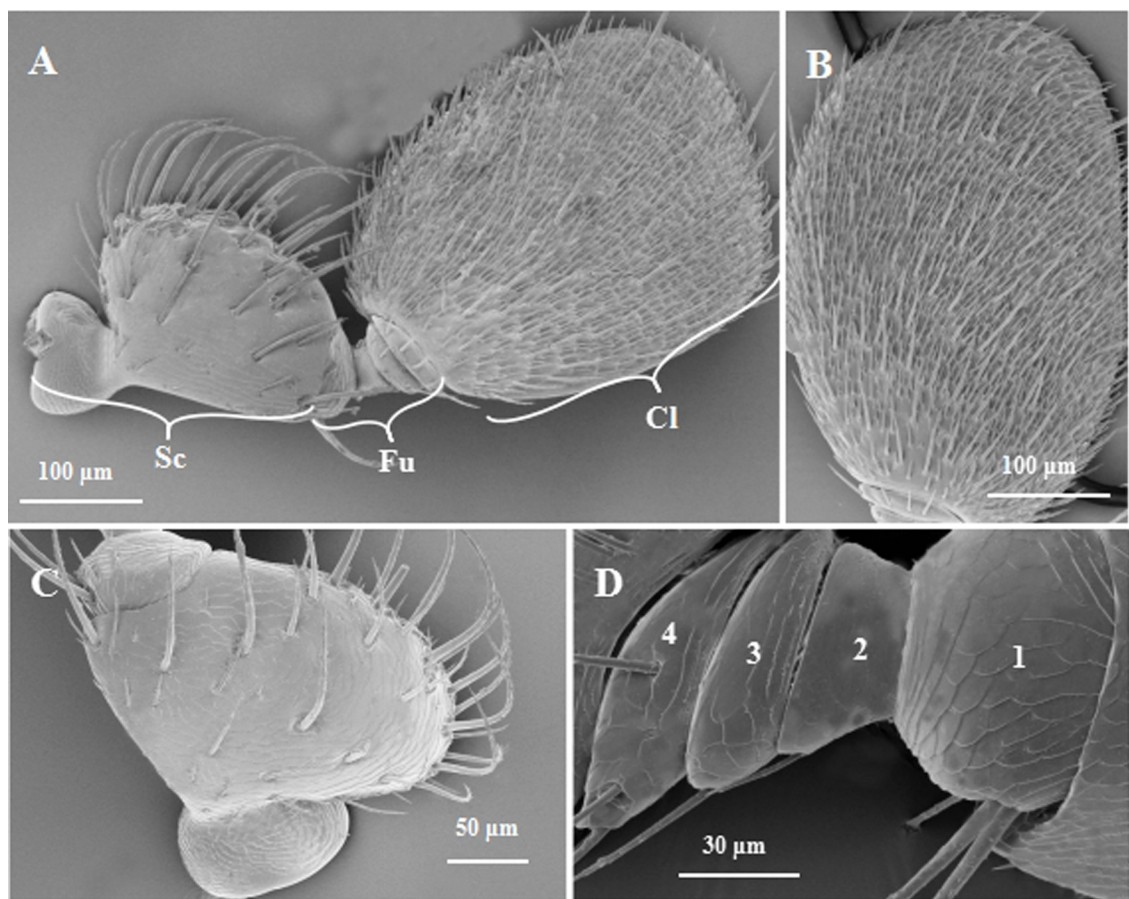

**Fig 2. Adult *Euplatypus parallelus* antennae in frontal vision.** A: Geniculated antennae of *E. parallelus*. B: Club. C: Scape. D: Funicle. Cl: Club; Fu: Funicle; Sc: Scape.

largest. Its number in the F4, F3 and F2 decreased successively (Table 4). The number of SChI appeared more larger in females than males (Table 4).

SChII was longer than SChI (Fig 3E). It was the longest sensillum in the antennae of *E. parallelus*, had a saw-toothed outer surface. There were about 21 spine-like branches in the wall of

**Table 1. Mean length and width of antennal segments in female and male *E. parallelus* (n = 10).**

| Antennal segments | | Length (µm) | | Width (µm) | |
|---|---|---|---|---|---|
| | | Female | Male | Female | Male |
| Scape | | 243.46 ± 1.43a | 235.54 ± 3.47a | 168.83 ± 2.52a | 159.21 ± 1.74b |
| Funicle | F1 | 72.45 ± 0.60a | 73.75 ± 0.65a | 86.34 ± 1.47a | 77.02 ± 1.63b |
| | F2 | 24.70 ± 0.88a | 25.71 ± 0.77a | 59.86 ± 1.25a | 50.44 ± 0.79b |
| | F3 | 18.05 ± 0.81a | 18.29 ± 0.55a | 77.36 ± 1.28a | 68.10 ± 0.99b |
| | F4 | 18.06 ± 0.77a | 19.18 ± 0.44a | 101.06 ± 1.59a | 92.19 ± 1.50b |
| | Pooled | 133.26 ± 1.58a | 136.92 ± 1.16a | —— | —— |
| Club | | 421.92 ± 4.88a | 362.78 ± 4.21b | 301.53 ± 3.05a | 285.23 ± 2.57b |
| Pooled | | 763.45 ± 5.40a | 698.96 ± 5.36b | —— | —— |

Date are presented as mean ± SE. Means in the same row followed by same letter are not significantly different ($P > 0.05$).

**Table 2. Morphological characteristics of sensilla present in female and male *E. parallelus* (n = 12).**

| Types of sensilla | Sex | Morphological characteristics | | | | | |
|---|---|---|---|---|---|---|---|
| | | Length (μm) | Width (μm) | Number of tooth | Tip | Wall | Shape |
| STI | Female | 17.93 ± 0.55a | 2.17 ± 0.06a | —— | Slightly sharp | Multiporous | Straight or curved |
| | Male | 18.39 ± 0.29a | 2.41 ± 0.06b | —— | | | |
| STII | Female | 22.11 ± 0.53a | 2.22 ± 0.05a | —— | Slightly sharp | Multiporous | Straight or curved |
| | Male | 22.33 ± 0.42a | 2.31 ± 0.05a | —— | | | |
| STIII | Female | 18.35 ± 0.67a | 1.98 ± 0.05a | —— | Sharp | Multiporous | Straight or curved |
| | Male | 18.57 ± 0.74a | 2.06 ± 0.04a | —— | | | |
| SChI | Female | 40.81 ± 1.61a | 2.82 ± 0.10a | 4.83 ± 0.27a | Sharp | Saw-tooth | Straight or curved |
| | Male | 42.28 ± 2.65a | 3.03 ± 0.13a | 5.67 ± 0.51a | | | |
| SChII | Female | 112.97 ± 5.22a | 4.38 ± 0.31a | 20.42 ± 0.96a | Sharp | Saw-tooth | Curved |
| | Male | 111.86 ± 3.01a | 5.57 ± 0.09b | 21.07 ± 0.88a | | | |
| SChIII | Female | 10.25 ± 0.53a | 1.28 ± 0.04a | —— | Sharp | Longitudinal furrows | Straight |
| | Male | 9.48 ± 0.55a | 1.39 ± 0.04a | —— | | | |
| SBI | Female | 11.82 ± 0.37a | 1.94 ± 0.03a | —— | Blunt | Multiporous | Straight |
| | Male | 12.29 ± 0.19a | 1.81 ± 0.04b | —— | | | |
| SBII | Female | 3.73 ± 0.10a | 2.21 ± 0.04a | —— | Pore | Smooth | Straight or curved |
| | Male | 3.86 ± 0.12a | 1.99 ± 0.03a | —— | | | |
| SBIII | Female | 11.78 ± 0.72a | 1.81 ± 0.04a | —— | Blunt | Smooth | Straight |
| | Male | 13.26 ± 0.27a | 1.91 ± 0.03a | —— | | | |
| SBIV | Female | 11.16 ± 0.28a | 1.42 ± 0.04a | —— | Blunt | Multiporous | Straight |
| | Male | 11.93 ± 0.20b | 1.42 ± 0.04a | —— | | | |
| SCo | Female | 7.57 ± 0.23a | 1.97 ± 0.07a | —— | Tapered Point | Grooved | Straight |
| | Male | 6.84 ± 0.72b | 2.05 ± 0.16a | —— | | | |
| SP | Female | —— | 0.42 ± 0.02a | —— | —— | —— | —— |
| | Male | —— | 0.49 ± 0.06b | —— | | | |

Date are presented as mean ± SE. Means in the same column followed by same letter on female and male are not significantly different *(P > 0.05)*.

this type; they were distributed alternately on the surface of the sensillum. Pores were not observed on the surface. SChII was distributed in the scape and F1. There were fewer SChII than SChI (Table 4). The number of SChII appeared more larger in females than males (Table 4).

SChIII was shaped like a needle (Fig 3F). It was perpendicular to the surface of the antennae, short and with a sharp tip. The surface of the sensillum was smooth, without pores and had no other accessory structures. It was distributed in the scape and F1. Its number was lower than SChII. There was no significant difference between males and females in the number of SChIII (Table 4).

Sensilla basiconica were straight or slightly curved, shaped like an awl and with a blunt tip. They were distributed dispersedly in the antennal club. Based on their morphological and size

**Table 3. The number of sensilla in antennal segments of female and male *E. parallelus* (n = 8).**

| Sex | Scape | F1 | F2 | F3 | F4 | Club | Pooled |
|---|---|---|---|---|---|---|---|
| Female | 80 ± 2a | 7 ± 1a | 1 ± 0a | 7 ± 1a | 8 ± 1a | 3272 ± 69a | 3374 ± 68a |
| Male | 70 ± 2b | 7 ± 1a | 1 ± 0a | 4 ± 1b | 7 ± 1a | 2858 ± 60b | 2948 ± 62b |

Date are presented as mean ± SE. Means in the same column followed by same letter are not significantly different *(P > 0.05)*.

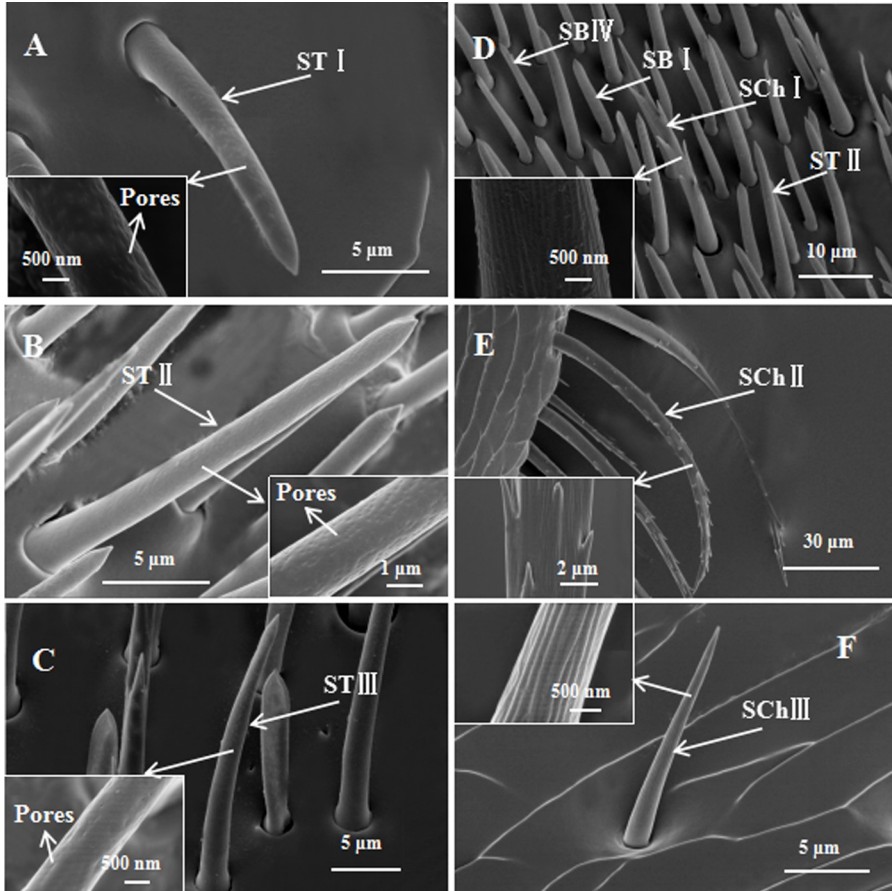

**Fig 3. Scanning electron micrographs of sensilla external cuticular structures on *Euplatypus parallelu* antennae.**
A: STI. Inset: The high magnification picture of STI. B: STII. Inset: The high magnification picture of STII. C: STIII.
Inset: The high magnification picture of STIII. D: SChI, SBI, SBIV, STII. Inset: The high magnification picture of SChI.
E: SChII. Inset: The high magnification picture of SChII. F: SChIII. Inset: The high magnification picture of SChIII.

features, we distinguished four subtypes of SB: I, II, III and IV. SBI was similar to STI, but it was shorter and thinner than STI (Fig 4A). Its wall was multiporous, and its number was more larger in females than males (Table 4).

SBII was straight or slightly curved, its wall was smooth and without pores, but it had a pores at the apex, and it was sunk in a deep socket (Fig 4B). SBII was the shortest sensillum in *E. parallelus*. It was scarce and distributed randomly in the club (Table 4).

SBIII was straight and looks like SBI, but its wall was smooth and without pores; it was distributed at the distal margin of the antennal club (Fig 4C). The number of SBIII was more larger in females than males.

Regarding morphology, SBIV was similar although finer than SBI (Fig 4D). This type was the most abundant structure on the antennae. *E. parallelus* females had a slightly higher number of SBIV than males (Table 4).

Sensilla coeloconica had a peg-like shape and a flame-shaped tip (Fig 5A). SCo was straight, not abundant and was distributed in the antennal club. Based on its morphological features, it was divided into an upper and a lower part. The lower half had a smooth cuticular surface, tapering gradually from the base to the tip. The upper half forms longitudinal grooves with obvious furrows by finger-shaped pegs, finishing with a rounded apex. The upper and lower

**Table 4. Distribution and number of sensilla present in female and male *E. parallelus* (n = 8).**

| Sensilla | Location | Females | Males |
|---|---|---|---|
| STI | Club | 443 ± 24a | 341 ± 25b |
| STII | Club | 544 ± 30a | 517 ± 37a |
| STIII | Club | 321 ± 10a | 333 ± 10a |
| ST | | 1311 ± 28a | 1190 ± 33b |
| SChI | Club | 173 ± 8a | 141 ± 5b |
| | F2 | 1 ± 0a | 1 ± 0a |
| | F3 | 7 ± 0a | 4 ± 0b |
| | F4 | 8 ± 0a | 7 ± 0a |
| | Pooled | 187 ± 8a | 153 ± 5b |
| SChII | Scape | 48 ± 2a | 39 ± 2b |
| | F1 | 5 ± 0a | 5 ± 0a |
| | Pooled | 52 ± 3a | 44 ± 2b |
| SChIII | Scape | 32 ± 1a | 31± 2a |
| | F1 | 3 ± 0a | 2± 0a |
| | Pooled | 33 ± 1a | 35 ± 2a |
| SCh | | 272 ± 8a | 231± 6b |
| SBI | Club | 708 ± 24a | 584± 21b |
| SBII | Club | 28 ± 2a | 24± 2a |
| SBIII | Club | 11 ± 1a | 7 ± 0b |
| SBIV | Club | 990 ± 26a | 856± 25b |
| SB | | 1737 ± 43a | 1470 ± 36b |
| SCo | Club | 54 ± 4a | 57 ± 5a |
| SP | Scape | 46 ± 1a | 42 ± 4a |
| | F1 | 2 ± 0a | 2 ± 0a |
| | F4 | 1 ± 0a | 0± 0a |
| | Club | 228 ± 25a | 266 ± 13a |
| | Pooled | 275 ± 24a | 311 ± 14a |

Date are presented as mean ± SE. Means in the same row followed by same letter are not significantly different (*P* > 0.05).

length ratio was close to 1:1. No significant difference in the number of SCo were found between males and females (Table 4).

Sensilla campaniforme was semispherical, shaped like a button, with a circle of smooth and clear-rim back wall (Fig 5C). It had a diameter of about 11 μm. In all FESEM photos, the SCa was found only in the two antennal scapes.

Sensilla furcatea was straight, with the base jointed within a socket (Fig 5B). Its wall was smooth and without furrows. There were fork-shaped branches at the distal end of this type. The angle of branches was small. This sensillum length was about 5.8 μm. It was scarce and distributed only in the proximal scape.

Sensory pits (SP) were circular concave pits that were distributed in the scape, F1, F4, and the club (Fig 5E). They were distributed sparsely in the funicle, but were abundant in the club. No significant difference in the number of SP were found between males and females (Table 4).

Squamifornia denticles (SD) were distributed mainly in the scape and funicle, and were attached to the surface of antennae (Fig 5D). The external cuticular structure of SD in males and females was similar.

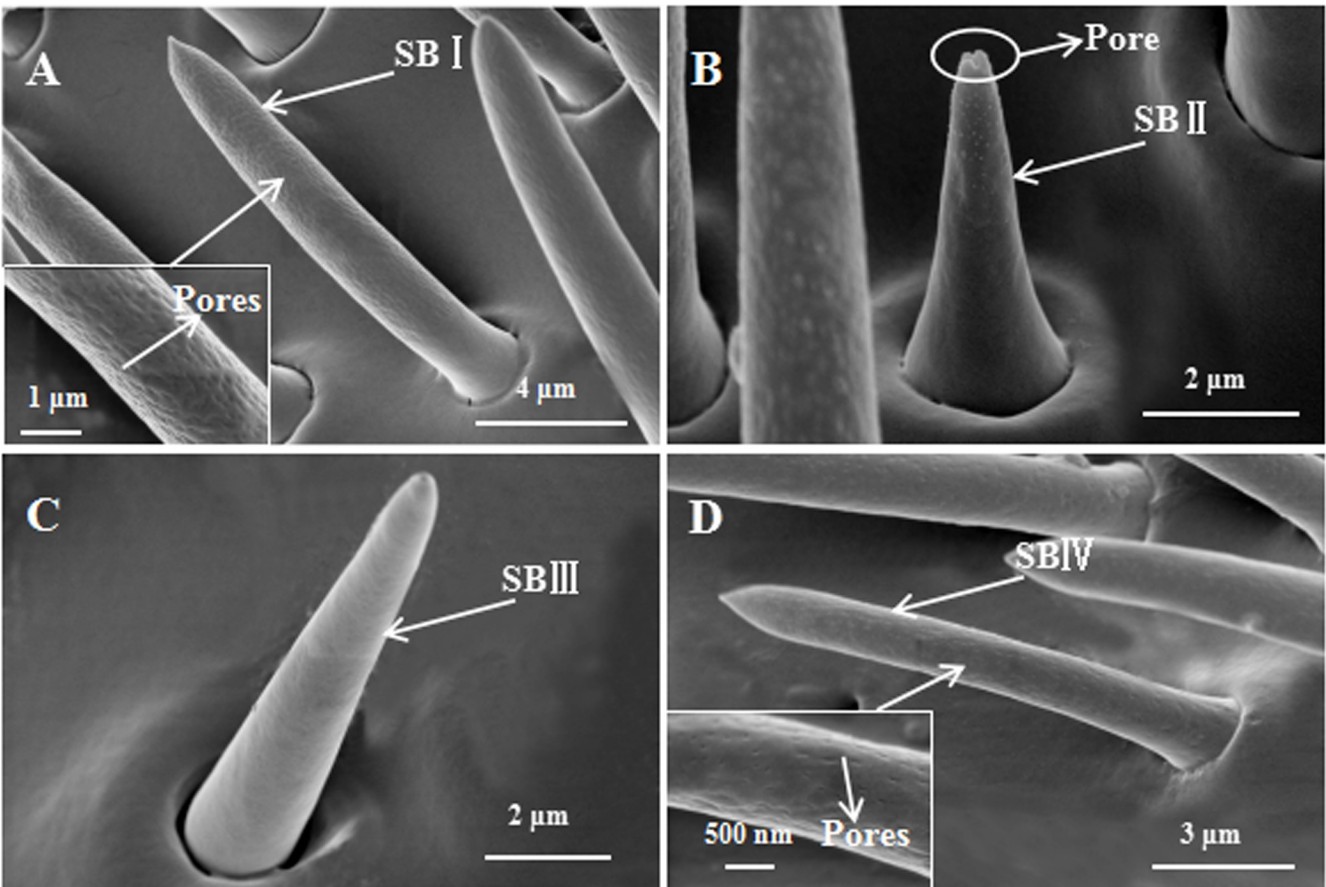

**Fig 4. Scanning electron micrographs of SB external cuticular structures on *Euplatypus parallelu* antennae.** A: SBI. Inset: The high magnification picture of SBI. B: SBII. C: SBIII. D: SBIV. Inset: The high magnification picture of SBIV.

SCo and SF were observed only on 1 or 2 antennae in all FESEM photos. They were excluded from statistical analyses.

## Discussion

The chemical communication system in *E. parallelus* is the key to its survival and reproduction. It mainly locates the host and find mates through plant-host volatiles and pheromones [21, 32]. Our research found that *E. parallelus* had geniculate antennae composed of 6 segments, namely the scape, 4-segmented funicle and club. Antennae of both sexes were morphologically similar. The length of antennae and club and the width of scape, all funicular antennomeres and club in females were significantly larger than those in males, corresponding to their body size difference. We did not observe any obvious sexual dimorphism with respect to type, morphology and distribution of sensilla. SChII, SChIII, SCa, and SF were distributed in the scape, whereas SChI and SChII were distributed in the funicle. Almost no sensilla were found in F2. There were 9 types (STI, STII, STIII, SChI, SBI, SBII, SBIII, SBIV, SCo), and the largest total number of sensilla was in the hammer head. It had been suggested that asymmetry in the distribution of sensilla on each segment of the antennae might be due to the peculiarities of their function, which would allow certain areas of the antennal surface to catch the signal molecules more effectively [33]. At present, the reports of antennal sensilla in bark beetles are

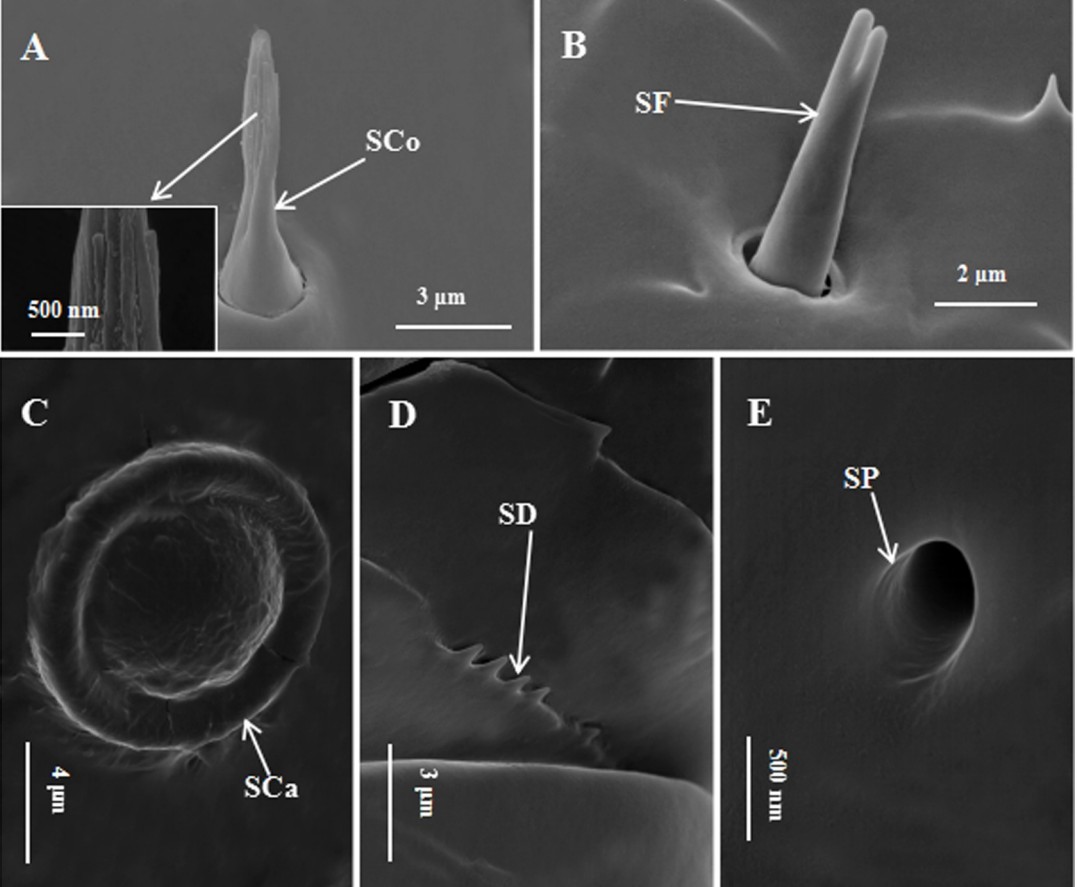

**Fig 5. Scanning electron micrographs of external cuticular structures on *Euplatypus parallelu* antennae.** A: SCo. Inset: The high magnification picture of SCo. B: SF. C: SCa. D: SD. E: SP.

mainly focused on Scolytinae, but there are few in Platypodinae. The 14 types of sensilla reported here are common for other bark beetles, but there is no unified system for naming them [34]; hence, we can compare various sensilla only according to their morphological structure.

ST in the antennae is only shorter than SChI and SChII. Pores were found in the wall of the three subtypes under FESEM, indicating an olfactory function. Similar structures were found in other bark beetles through morphological comparison. Among them, we found STI in *E. parallelus* was similar to that on the antennae of *Scolytus multististriatus* [35], *Ips acuminatus* [26] and *Xylosandrus compactus* [25]. Chen et al. [36] and Dickens and Payne [37] described, respectively, sensilla trichodea 3 in *Dendroctonus valens* and sensilla trichodea 2 in *Dendroctonus frontalis* that were similar to STII in *E. parallelus*. Similarly, the STIII described here in *E. parallelus* corresponded to sensilla trichodea 2 in *Tomicus yunnanensis*, *Tomicus minor* and *Tomicus brevipilosus* [31], also resembling sensilla trichodea 1 in *Xylosandrus germanus* and *Xylosandrus crassiusculus* [24]. STI has fewer pores in the wall than STII and STIII. Through extirpation experiments, Borden and Wood [38] found that ST functioned in the olfactory perception in *Ips confusus* and *Ips paraconfusus*. Moeck [39] observed two neurons in these sensilla and concluded a mechanosensory role was unlikely. Palma et al. [40] used transmission electron microscopy (TEM) to observe many pores in ST of *Hylastinus obscurus*, forming radial channels connecting the surface to the lumen. The olfactory function was considered

the most probable. Electrophysiological studies proved these sensilla responded to the phero-mones, but had a poor response to one single general odor [41, 42].

ST can function partly as mechanoreceptors [43]. ST found on *Dendroctonus vitei* were nonporous; their positions on the antennae suggested they might play a role in mechanorecep-tion [44]. Chen et al. [36] observed by TEM only a thin lumen in cross-section of ST in *Dendroctonus valens*, surrounded by thick cuticle and lacking wall pores, suggesting involvement in the sense of touch, also a possible function in sensing air flow rate, gravity and sound waves [20]. The number of STI of antennae was significantly larger in females than males (Table 4). These results indicated that, compared with males, female adults of *E. parallelus* needed more chemoreceptors to complete their life cycle.

SB were the most abundant structure on the antennae (Table 4); this type of sensilla was commonly seen in other bark beetles. Among them, SBI described in this study was similar to those on the antennae of *Scolytus multististriatus* [35], *Ips typographus* [45], *Dendroctonus valens* [36], and *Xyleborus saxese*ni [46]. SBII was similar in appearance to those described in *Xylosandrus germanus*, *Xylosandrus crassiusculus* [24], *Dendroctonus valens* and *Dendroctonus rhizophagus* [47]. SBIV resembled the sensilla basiconica 1 in *Dendroctonus valens* [36]. Mor-phologically, SBI and SBIV were multiporous chemosensilla with pitted surfaces, and SBII had a pore in the apical part, suggesting a chemoreceptor role for these sensilla in *E. parallelus*. Chen et al. [36] observed the numerous pores and branched dendrites in the TEM photos of SB. These structures were considered to be evidence that SB function as olfactory receptors [43, 48]. Their specific functions have been described in other bark beetles. Electrophysiologi-cal studies found that SB in *Dendroctonus frontalis* and *Dendroctonus ponderosae* responded to pheromone components and host-produced terpenes [37, 49, 50]. In *Ips confusus* a sensitivity of the SB to pheromones was supposed [38]. Therefore, we think SB may be involved in odor recognition, host location and discrimination of aggregated pheromones.

The wall of SBIII was smooth and without pores (Fig 4C); it was distributed sparsely at the distal margin of the antennal club. This type was similar in appearance to those described in *Xylosandrus compactus* [25] and *Dendroctonus vitei* [44]. Payne et al. [51] suggested that they may have chemoreceptor functions, but the chemical types to which they respond could be dif-ferent from those sensed by the multiporous SB. Dendrites might extend from the hair lumen to the tip and might sense $CO_2$, water vapor, or other chemicals [39]. Pores were observed in the morphology of six sensilla on the antennae of *E. parallelus* (Figs 3A–3C, 4A, 4B and 4D), suggesting olfaction as the likely function of these sensilla. They were all distributed on the club, with STI, SBI and SBIV being more numerous in female than male adults (Table 4). These studies may indicate a greater olfactory ability in female than male *E. parallelus*. ST with wall pores are present on the antennae of all insect species ever investigated, such as *Coleo-phora obducta* (Meyrick) (Lepidoptera: Coleophoridae) [52], *Pseudacteon tricuspis* (Diptera: Phoridae) [53], *Eupristina* sp. (Hymenoptera: Agaonidae) [54], *Stephanitis nashi* (Hemiptera: Tingidae) [55]. In many other beetles *Tetropium fuscum* (Fabr.) (Coleoptera: Cerambycidae) [56], *Tetrigus lewisi* Candèze (Coleoptera: Elateridae) [57], *Dastarcus helophoroides* (Fair-maire) (Coleoptera: Bothrideridae) [58], ST has been shown by electrophysiology to be contact pheromone receptor. In some moth species, it has been demonstrated that they function as sex pheromone receptors [30, 59]. Shields and Hildebrand showed that ST of the female *Manduca sexta* could respond to aromatic or terpenoid odorants [60]. In *Drosophila* (Diptera) antennae, ST functions as pheromone and plant volatiles receptors [61].

SCh were distributed in all segments of the antennae in *E. parallelus*. Based on their mor-phological structure, we distinguished three subtypes of SCh. SChI and SChII were nonporous. They had long longitudinal furrows and spine-like branches in the wall. They were the longest sensilla in the antennae of *E. parallelus*, and were considered likely to be mechanoreceptors.

SChI was similar in appearance to those on the antennae of *Hylastinus obscurus* [40], *Dendroctonus valens* [36], and *Ips typographus* [45]. SChII described here was similar to that in *Ips confusus* [38]. Moeck [39] found SCh were thick-walled and probably all innervated by a single neuron. When the antennae of bark beetles work, the long sensilla of the antennae were the first to contact the substrate, assisting the beetle to confirm the position [51]. The saw-toothed structures would function to detect and transmit diverse mechanical stimuli [22], or might also function as the wind velocity receptors [62]. They were comparatively long and wide, indicating SCh might provide some degree of protection over the shorter SB and SCo [63].

The surface of SChIII was smooth, without pores and had longitudinal furrows (Fig 3F). These sensilla were also referred to as "Böhm bristles", distributed on almost all Coleoptera insects. Wang et al. [31] revealed by TEM these sensilla were devoid of wall pores, suggesting a non-olfactory role, and speculating they were gravity receptors. They were able to buffer gravity when encountering mechanical stimuli [20]. The number of SChI and SChII was significantly larger in females than males (Table 4). These results showed that female adults could feel mechanical stimulation by using the frontal side of their antennae, which is consistent with the conclusion that the female adults are the main force during the gallery construction.

SCo were scarce and distributed randomly in the antennal club; they were found in almost all bark beetles, such as *Xylosandrus germanus*, *Xylosandrus crassiusculus* [24], *Hylastinus obscurus* [40], *Ips typographus* [45], and *Dendroctonus valens* [36]. Although the nomenclature of these sensilla is not entirely consistent, they were completely identical in morphology to SCo in *E. parallelus*. No significant difference in the number of SCo were found between males and females, suggesting that SCo have a similar function in both sexes. Whitehead [23] characterized SCo as multiporous sensilla with deep longitudinal grooves (MPG) and innervated by four neurons [64]. MPG were related to thermo-chemical and thermo-hygro receptors. Some studies also suggested that MPG increased the sensilla surface area to accept more odor molecules. It is generally assumed that sensilla with such morphology would exhibit chemosensory functions, including thermo-chemical [65] and thermo-hygro reception [48] and olfactory function [43].

SCa was barely found on the antennae of *E. parallelus*, and was situated only in the two antennal scapes in all FESEM photos. Whitehead [23] and Moeck [39] found the same structure in *Dendroctonus ponderosae* and *Trypodendron lineatum*, respectively. The sensilla were generally considered to be proprioceptor that could sense the stresses in the cuticle resulting from mechanical deformation, responding immediately to changes in the cuticular system [66].

SF was reported in *Tomicus yunnanensis* [31]; they were furcated at the tip and had a smooth surface (Fig 5B). They were scarce and distributed only in the proximal scape. On the basis of morphology and distribution, they might have the same roles as SChIII. We assumed SF was a morphological variant of SChIII.

Beside the sensory organs, sensory pits were also observed on the surface of the antennae. No significant difference in the number of SP were found between males and females. This structure was also found in other bark beetles, such as *Dendroctonus valens* [47], *Tomicus yunnanensis* [67], *Xylosandrus compactus* [25], and *Ips acuminatus* [26]. However, its function is unknown. In some insects, SP might degrade molecules of pheromones or plant–host volatiles to prevent them overloading the antennal chemosensilla [68–71], and may also play a role in secreting demulcent, hormone, lubricant and other substances [72, 73].

In this study, we described and analyzed the morphology, structure, distribution, and quantity of the antennal sensilla in *E. parallelus* using FESEM. We speculated the functions of various sensilla and compared our findings with the published reports. In the future, it will be necessary to clarify the functions of various sensors in insect behavior by conjunction with TEM and electrophysiology. In addition, insects can rely on multiple organs to sense information. Some reports have indicated that other sensilla with a chemoreceptive function are

present on the mouthparts, ovipositor and tarsi. Therefore, a study of the sensory equipment in different organs to clarify the relationship between chemical receptors and behavior mechanisms of *E. parallelus*.

## Supporting information

**S1 Data.**
(XLS)

## Acknowledgments

We are grateful to Ms Jingyun Zhou from the Instrumental Analysis and Research Center of South China Agricultural University for her assistance with SEM. We thank Ms Junyu Chen (Environment and Plant Protection Institute, Chinese Academy of Tropical Agricultural Science, Haikou, China) for her comments.

## Author Contributions

**Conceptualization:** Wei-Jian Wu.

**Data curation:** Gaoke Lei, Wei-Jian Wu.

**Formal analysis:** Yueguan Fu, Wei-Jian Wu.

**Investigation:** Gaoke Lei, Wei-Jian Wu.

**Methodology:** Wei-Jian Wu.

**Resources:** Yueguan Fu, Wei-Jian Wu.

**Writing – original draft:** Gaoke Lei.

**Writing – review & editing:** Yueguan Fu, Wei-Jian Wu.

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
