## [Decision Letter · Decision Letter 0]

22 Jul 2020

PONE-D-20-18269

Type and distribution of sensilla in the antennae of Euplatypus parallelus (F.) (Coleoptera: Curculionidea, Platypodidae)

PLOS ONE

Dear Dr. Wu,

Thank you for submitting your manuscript to PLOS ONE. After careful consideration, we feel that it has merit but does not fully meet PLOS ONE’s publication criteria as it currently stands. Therefore, we invite you to submit a revised version of the manuscript that addresses the points raised during the review process.

This is a nice description of  the morphology and the external organization of the antennae of beetle species belonging to a poorly studied group. Note that “ultrastructure” (abstract line 42) is usually used for a description at cellular and infra cellular levels and is not suitable for SEM observations of external cuticular structures. The data are worth to be published. However, the manuscript requires editing for English usage, sentence construction, and grammar. This is a sample of sentences needing revision: line 95 “the cutting end “; line 102:  “… and placed antennae…”; line 110 “antennal data were measured”, data are not measured but the output of measurement; line 142 “Sensilla trichodea: they are trichoid sensilla”. Line 54 “Sensilla chaetica: the shape like a thorn” a verb is missing. Line 231 “Among them,STⅠ we found in E. parallelus was similar to that…”. Etc. Some sentences need also revision because of unreliable statements: Line 68-69. I am not convinced that small size and capacity of propagation are a limit to chemical control. Insecticides are efficient against aphids or thrips for instance and the problems come from resistance. Line 93: the scientific name of the tree is missingLines 216-217 “The larger antennae of female adults contained more sensilla and performed better in some functions.” You did not perform a functional study so performance of the female antennae cannot be evaluated.

We look forward to receiving your revised manuscript.

Kind regards,

Michel Renou, Ph.D

Academic Editor

PLOS ONE

Journal Requirements:

https://www.sciencedirect.com/science/article/abs/pii/S0968432819303002?via%3Dihub

In your revision ensure you cite all your sources (including your own works), and quote or rephrase any duplicated text outside the methods section. Further consideration is dependent on these concerns being addressed.

4. Please ensure that you refer to Figure 2 in your text as, if accepted, production will need this reference to link the reader to the figure.

5. Please include your tables as part of your main manuscript and remove the individual files. Please note that supplementary tables (should remain/ be uploaded) as separate "supporting information" files

6. We note you have included a table to which you do not refer in the text of your manuscript. Please ensure that you refer to Tables 2 & 4 in your text; if accepted, production will need this reference to link the reader to the Table.

Reviewers' comments:

Reviewer's Responses to Questions

**Comments to the Author**

1. Is the manuscript technically sound, and do the data support the conclusions?

Reviewer #1: Partly

2. Has the statistical analysis been performed appropriately and rigorously? 

Reviewer #1: Yes

3. Have the authors made all data underlying the findings in their manuscript fully available?

Reviewer #1: No

4. Is the manuscript presented in an intelligible fashion and written in standard English?

Reviewer #1: No

5. Review Comments to the Author

Reviewer #1: The authors have done a good job of describing the sensillae on the antenna of the ambrosia beetle Euplatypus parallelus, a desctructive invasive species. However, additional information is needed to support several conclusions. 1) Support for whether SP is a secretory gland or has underlying sensory cells requires either transmission electron microscopy (TEM) of thin sections or SEM imaging of KOH digested thick sections (Bin and Vinson, Int. J. Insect Morphol. and Embryol. 15: 129–138, 1986). 2) Assessment of whether the depressions observed on the surface of sensillae are pores requires either TEM of thin sections or silver staining (Navasero & Elzen, Proc. Ent. Soc. Wash. 93, 737–747, 1991).

The manuscript contains many grammatical errors that must be corrected before the paper can be published. Here are the errors I found in just the first 100 lines:

42 ultratructure  ultrastructure

61 infect  infecting

83 morphology Scolytinae  morphology of Scolytinae

85 , this  . This

95 cutting  cut

96 the cages  cages

96-97 and collected the adult beetles emerging from the gallery.  and the adult beetles emerging from the gallery were collected.

98 , the  . The

6. PLOS authors have the option to publish the peer review history of their article (what does this mean?). If published, this will include your full peer review and any attached files.

Reviewer #1: **Yes: **Robert Renthal

---

## [Author Response · Author response to Decision Letter 0]

13 Aug 2020

Dear editors and reviewers,

Thank you for your letter and for the reviewers’ comments concerning our manuscript entitled Type and distribution of sensilla in the antennae of Euplatypus parallelus (F.) (Coleoptera: Curculionidea, Platypodidae). Those comments are all valuable and very helpful for revising and improving our paper, as well as the important guiding significance to our researches. We have studied comments carefully and have made the correction, which we hope to meet with approval. The main corrections in the paper and the responds to the reviewer’s comments are as following:

Editor comments:

1.“Ultrastructure” (abstract line 42) is usually used for a description at cellular and infra cellular levels and is not suitable for SEM observations of external cuticular structures.

2.The manuscript requires editing for English usage, sentence construction, and grammar.

3.Line 68-69, not convinced that small size and capacity of propagation are a limit to chemical control.

4.Line 93, the scientific name of the tree is missing.

5.Lines 216-217, you did not perform a functional study so performance of the female antennae cannot be evaluated.

Response:

1.Thanks for the referee’s kind suggestion. We are really sorry for the errors in the use of terminology. We refer to the editors’ suggestion and revise “ultrastructure” to “external cuticular structures”.

2.Thanks for the referee’s kind suggestion. We are very sorry for our mistakes in English usage, sentence construction, and grammar. We entrusted a professional scientific editing service that edited our manuscript. we would like to express their gratitude to EditSprings (https://www.editsprings.com/) for the expert linguistic services provided. We provided the Editorial certificate in supporting information.

3.Thanks for the referee’s kind suggestion. It is our negligence and we are sorry about “E. parallelus is difficult to control with pesticides because this beetle is small and propagates rapidly”. Depending on comments, related content has been improved, we add “it live hidden away” in the manuscript.

4.We add the scientific name of the tree, Hevea brasiliensis.

5.We agree with the reviewer such you did not perform a functional study so performance of the female antennae cannot be evaluated. Our research is certainly not enough to explain this problem. We have deleted this sentence.

Reviewer comments:

1.Support for whether SP is a secretory gland or has underlying sensory cells requires either transmission electron microscopy (TEM) of thin sections or SEM imaging of KOH digested thick sections.

2.Assessment of whether the depressions observed on the surface of sensillae are pores requires either TEM of thin sections or silver staining.

3.The manuscript contains many grammatical errors that must be corrected before the paper can be published.

Response:

1.We agree entirely with this suggestion by the reviewer. Our research is certainly not enough to explain this problem. We have deleted this sentence.

1.Acknowledge the reviewer for his valuable suggestions. Our research is certainly not enough to explain this problem. We use either a phylogenetic framework (comparative morphology among species which same in morphology) to discover something new. In the discussion, we inferred their functions by comparing the external cuticular structures of different bark beetles.

2.Thanks for the referee’s kind suggestion. We entrusted a professional scientific editing service that edited our manuscript.

Attachments

We have deposited our laboratory protocols in protocols.io: http://dx.doi.org/10.17504/protocols.io.bjjkkkkw

Additional requirements:

1.We ensured that our manuscript meets PLOS ONE's style requirements.

2.In our revision, we addressed some minor overlapping text, and we ensure we cite all our sources.

3.We are very sorry for our mistakes in English usage, sentence construction, and grammar. We entrusted a professional scientific editing service that edited our manuscript.

4.We refer to Figure 2 in our text.

5.We supplemented our tables as part of our main manuscript and remove the individual files.

6.We refer to Tables 2 & 4 in our text.

We have uploaded our figure files to the Preflight Analysis and Conversion Engine (PACE) digital diagnostic tool, the details can be found in file “Figure”.

We would like to make changes to our financial disclosure: 

This study was supported by the Special Fund for Agro-scientific Research in the Public Interest of China (201103026-4) and the Earmarked Fund for China Agriculture Research System (CARS-33-BC2). The funders had no role in study design, data collection and analysis, decision to publish, or preparation of the manuscript.

We tried our best to improve the manuscript and made some changes in the manuscript.  These changes will not influence the content and framework of the paper. And here we did not list the changes but marked in revised paper.

Many grammatical or typographical errors have been revised.

Thank you and all the reviewers for the kind advice.

Special thanks to you for your good comments.

Sincerely yours, 

G.K. Lei

Corresponding author:

Name: W.J. Wu

Laboratory of Insect Ecology

South China Agricultural University

Guangzhou, 510642

Guangdong, P. R., China

E-mail: weijwu@scau.edu.cn

---

## [Editor Report · Decision Letter 1]

19 Aug 2020

PONE-D-20-18269R1

Type and distribution of sensilla in the antennae of <euplatypus parallelus=""> (F.) (Coleoptera: Curculionidea, Platypodidae)

PLOS ONE

Dear Dr. Wu,

Thank you for submitting your manuscript to PLOS ONE. After careful consideration, we feel that it does not meet PLOS ONE’s publication criteria as it currently stands. The writing still need to be improved in order to better formulate the scientific hypothess and to make the desciption of cuticular structures, the comparisons between species and the speculation about function, clear, informative, and precise.  I have also noted misuse of terms and grammar and syntax errors that were not fixed by the linguistic revision. Please, download the annotated pdf for my specific comments that I hope could be helpful, but are not exhaustive.</euplatypus>

<euplatypus parallelus=""></euplatypus>

We look forward to receiving your revised manuscript.

Kind regards,

Michel Renou, Ph.D

Academic Editor

PLOS ONE

---

## [Author Response · Author response to Decision Letter 1]

2 Oct 2020

Dear editors,

Thank you for your letter and for the comments concerning our manuscript entitled Type and distribution of sensilla in the antennae of Euplatypus parallelus (F.) (Coleoptera: Curculionidea, Platypodidae). Those comments are all valuable and very helpful for revising and improving our paper, as well as the important guiding significance to our researches. We have studied comments carefully and have made the correction, which we hope to meet with approval. The main corrections in the paper and the responds to the comments are as following:

Editor comments:

1.The writing still need to be improved in order to better formulate the scientific hypothess and to make the desciption of cuticular structures, the comparisons between species and the speculation about function, clear, informative, and precise. 

2.Misuse of terms and grammar and syntax errors that were not fixed by the linguistic revision.

Response:

1.Thanks for the referee’s kind suggestion. We added the following (325-335) to the discussion to better formulate the scientific hypothess and to make the desciption of cuticular structures, the comparisons between species and the speculation about function, clear, informative, and precise. 

ST with wall pores are present on the antennae of all insect species ever investigated, such as Coleophora obducta (Meyrick) (Lepidoptera: Coleophoridae), Pseudacteon tricuspis (Diptera: Phoridae), Eupristina sp. (Hymenoptera: Agaonidae), Stephanitis nashi (Hemiptera: Tingidae). In many other beetles Tetropium fuscum (Fabr.) (Coleoptera: Cerambycidae), Tetrigus lewisi Candèze (Coleoptera: Elateridae), Dastarcus helophoroides (Fairmaire) (Coleoptera: Bothrideridae), ST has been shown by electrophysiology to be contact pheromone receptor. In some moth species, it has been demonstrated that they function as sex pheromone receptors. Shields and Hildebrand showed that ST of the female Manduca sexta could respond to aromatic or terpenoid odorants. In Drosophila (Diptera) antennae, ST functions as pheromone and plant volatiles receptors.

2.Thanks for the referee’s kind suggestion. We are very sorry for our mistakes in English usage, sentence construction, and grammar. We refer to the editors’ suggestion and revise them in new manuscript. 

We would like to make changes to our financial disclosure: 

This study was supported by the Special Fund for Agro-scientific Research in the Public Interest of China (201103026-4) and the Earmarked Fund for China Agriculture Research System (CARS-33-BC2). The funders had no role in study design, data collection and analysis, decision to publish, or preparation of the manuscript.

We tried our best to improve the manuscript and made some changes in the manuscript.  These changes will not influence the content and framework of the paper. And here we did not list the changes but marked in revised paper.

Many grammatical or typographical errors have been revised.

Thank you and all the reviewers for the kind advice.

Special thanks to you for your good comments.

Sincerely yours, 

G.K. Lei

Corresponding author:

Name: W.J. Wu

Laboratory of Insect Ecology

South China Agricultural University

Guangzhou, 510642

Guangdong, P. R., China

E-mail: weijwu@scau.edu.cn

---

## [Editor Report · Decision Letter 2]

13 Oct 2020

Type and distribution of sensilla in the antennae of <euplatypus parallelus=""> (F.) (Coleoptera: Curculionidea, Platypodidae)

PONE-D-20-18269R2</euplatypus>

Dear Dr. Wu,

We’re pleased to inform you that your manuscript has been judged scientifically suitable for publication and will be formally accepted for publication once it meets all outstanding technical requirements.

Kind regards,

Michel Renou, Ph.D

Academic Editor

PLOS ONE
---

## [Editor Report · Acceptance letter]

15 Oct 2020

PONE-D-20-18269R2 

Type and distribution of sensilla in the antennae of *Euplatypus parallelus* (F.) (Coleoptera: Curculionidea, Platypodinae) 

Dear Dr. Wu:

I'm pleased to inform you that your manuscript has been deemed suitable for publication in PLOS ONE. Congratulations! Your manuscript is now with our production department. 

Kind regards, 

on behalf of

Dr Michel Renou 

Academic Editor

PLOS ONE